# Coronary embolism in left-sided infective endocarditis. A retrospective analysis from a high-volume surgical centre and review of the literature

Ruggero Mazzotta[1,2], Matteo Orlandi[1,2]*, Valentina Scheggi[3], Niccolò Marchionni[1,2], Pierluigi Stefàno[2,4]

1 Division of General Cardiology, Cardiothoracovascular Department, Careggi University Hospital, Florence, Italy, 2 Department of Experimental and Clinical Medicine, University of Florence, Florence, Italy, 3 Division of Cardiovascular and Perioperative Medicine, Cardiothoracovascular Department, Careggi University Hospital, Florence, Italy, 4 Division of Cardiac Surgery, Cardiothoracovascular Department, Careggi University Hospital, Florence, Italy

* matteo.orlandi@unifi.it

## Abstract

### Background

Septic embolization is a common and potentially life-threatening complication of infective endocarditis (IE), with a prevalence of 22–50%. While acute coronary syndrome secondary to septic embolism is rare, it poses significant risks.

### Aims

This study examines coronary embolism (CE) in left-sided IE, describing clinical characteristics and outcomes.

### Methods

We retrospectively analysed 649 patients with non-device-related left-sided IE treated between January 2013 and December 2023 in a high-volume surgical centre. CE was diagnosed via ECG, clinical and laboratory signs of acute coronary syndrome, and confirmed by coronary angiography or magnetic resonance imaging. All patients were treated according to current European Society of Cardiology guidelines. A structured follow-up was performed.

### Results

Among patients included in the study, surgery was performed in 514 (79%) patients. Median follow-up duration was 4.7 years. CE occurred in 8 (1.2%) patients, and 6 (80%) of them were treated surgically. We found no significant differences in overall mortality rate between patients with or without CE (p = 0.65). Finally, cerebral embolism was significantly more frequent in patients with than without CE (75% vs 25%, p = 0.006, post-hoc power 87.8%).

**Data Availability Statement:** All relevant data are within the manuscript.

**Funding:** The author(s) received no specific funding for this work.

**Competing interests:** The authors have declared that no competing interests exist.

## Conclusion

CE is a rare but severe complication of IE, significantly associated with cerebral embolism. Early recognition and treatment are crucial to improve patient outcomes. Multicentre studies with larger patient populations are needed to further elucidate risk factors and enhance prognosis for CE in IE patients.

## Introduction

Septic embolization is a frequent and potentially life-threatening complication of infective endocarditis (IE) [1, 2]. Embolic risk is high among patient diagnosed with IE, with a prevalence of 22–50%, and most emboli involve the central nervous system [3]. Acute coronary syndrome (ACS) secondary to septic embolism is rare and has been addressed by few studies, the oldest ones reporting coronary arteries microemboli in more than 60% of post-mortem examinations [4, 5]. ACS in the setting of IE may not only be secondary to coronary embolism (CE). In fact, several studies found that the most common cause of ACS during IE is compression of a coronary artery by periaortic pseudoaneurysm or abscess [6–10]. Other less common mechanisms include obstruction of the coronary ostium by a large vegetation, especially in case of fungal endocarditis, [11] coexistent coronary artery disease that emerges clinically during the infection due to anaemia, fever, and activation of the coagulation system, and the occurrence of severe aortic valve regurgitation, which can per se cause myocardial ischemia because of reduced coronary perfusion pressure [6].

## Methods

### Patient selection

We retrospectively analysed 649 consecutive patients with non-device-related left-sided IE admitted to our high-volume surgical centre between January 2013 and December 2023. The study was approved by the local ethics Committee that, in keeping with statements by the Italian Regulatory Authorities for retrospective, observational studies, [12] granted a waiver of informed consent from study participants. Data collection took place between January 2019 and July 2024. Data for analysis were obtained from electronic hospital charts, anonymized, and protected by password. CE was diagnosed by ECG, clinical and laboratory signs of ACS and confirmed by coronary angiography or magnetic resonance imaging (MRI).

### Diagnostic work-up

We followed current European Society of Cardiology guidelines for diagnostic work-up and treatment strategies [13]. Accordingly, three sets of blood cultures were collected on admission and transoesophageal echocardiography (TEE) was performed in all patients for diagnosis confirmation. Systemic embolism was sought clinically and radiologically by brain and chest computed tomography (CT) plus abdominal CT or ultrasound scan. All patients were treated with empirical and, whenever feasible, targeted antimicrobial therapy.

### Surgical indication and operative technique

An Endocarditis Team including a cardiac surgeon, a cardiologist, an anaesthesiologist, an internist, a neurologist and an infectious disease specialist evaluated each patient. Surgical risk

was estimated by EuroSCORE II [14]. Surgery was advised according to current European Society of Cardiology guidelines [13].

### Follow-up and study endpoints

The duration of follow-up was calculated from the time of IE diagnosis. A structured phone interview was implemented to update the follow-up to December 2023. Our primary objective was describing clinical characteristics and the outcome of patients with CE.

### Statistical analysis

The chi-square and the Mann-Whitney or Kruskal-Wallis tests were used to compare respectively proportions and continuous variables with normal or non-normal distribution. The Kaplan-Meier method was used to estimate the survival probability. All tests were 2-sided, and statistical significance was defined as a p value $<0.05$. All analyses were performed with SPSS software version 24.0 (IBM Corp., Armonk, New York).

## Results

The demographic, echocardiographic, clinical and microbiological characteristics of the 649 patients are shown in Table 1. Surgery was not indicated in 68 (10%) out of 649 patients, whereas 67 (10%) were not operated because of prohibitive surgical risk; the remaining 514 (80%) underwent surgical valve repair (n = 93, 18%) or replacement (n = 421, 82%). The median follow-up duration was 4.7 years. CE occurred in 8 (1.2%) patients. The main clinical characteristics of the 8 patients with CE are summarized in Table 2. 6 patients (80%) with CE were treated surgically. We found no significant differences in overall mortality rate between patients with or without CE (p = 0.65). Three-year all-cause mortality rate of the whole cohort was 35%. Despite the small sample of CE patients, we found a statistically significant association of coronary and cerebral embolism, with a post-hoc power of 87.8%. All statistical analyses were limited by the small sample of CE patients.

### Clinical presentation and treatment of the 8 patients with CE

**Case 1.** A 52-year-old man presented to the emergency department complaining fever (38.5°C) and myalgias for one week. His past medical history reported a Bentall operation with implantation of a biologic prosthetic valve 14 years before. Shortly after hospital admission, he had typical chest pain: the ECG showed inferior ST-elevation. Transthoracic echocardiography revealed left ventricular inferior apical akinesia and a non-echogenic image suggestive of a periaortic abscess. Immediate coronary angiography revealed an embolic occlusion of mid-distal left circumflex artery, treated with thromboaspiration (Fig 1). Empirical antibiotic therapy with vancomycin, gentamicin and rifampicin was started. Blood cultures tested positive for *Staphylococcus lugdunensis*, and antimicrobial therapy was switched to oxacillin. The patient underwent redo Bentall surgery with a mechanical prosthetic valve. The postoperative course was uneventful and at discharge the aortic prosthesis was well-functioning.

**Case 2.** An 87-year-old woman was admitted to the emergency department for typical chest pain, fever (37.9°C) and general malaise. An ECG showed an inferior and lateral ST-elevation. At coronary angiography, there was an embolic occlusion of distal left anterior descending artery. No interventional procedure was performed. The patient underwent TEE with evidence of severe calcific mitral valve stenosis associated with moderate regurgitation and findings suggestive of anterior leaflet vegetations. Blood cultures were negative. Empirical antibiotic therapy (amoxicillin/clavulanate and gentamicin) was started. During her hospital

**Table 1. Demographic, echocardiographic, clinical and microbiological characteristics of 649 patients with left-sided infective endocarditis by the occurrence of coronary embolism.**

| | Coronary embolism | | p-value |
|---|---|---|---|
| | No (n = 641) | Yes (n = 8) | |
| Age, years, mean ± SD | 67.0 ± 14.0 | 53.3 ± 23.8 | 0.05 |
| Female sex, n (%) | 220 (34.3) | 2 (25.0) | 0.58 |
| BMI, mean ± SD | 24.8 ± 4.3 | 23.6 ± 4.0 | 0.7 |
| Chronic kidney disease, n (%) | 151 (23.6) | 3 (37.5) | 0.35 |
| Diabetes, n (%) | 136 (21.2) | 0 (0) | 0.14 |
| Hypertension, n (%) | 403 (62.9) | 2 (25.0) | 0.058 |
| Dyslipidaemia, n (%) | 198 (30.9) | 1 (12.5) | 0.22 |
| Intravenous drug abuse, n (%) | 40 (6.2) | 0 (0) | 0.46 |
| Native valve, n (%) | 389 (60.7) | 3 (37.5) | 0.18 |
| Aortic valve IE, n (%) | 362 (56.5) | 5 (62.5) | 0.86 |
| Perivalvular extension, n (%) | 133 (20.7) | 3 (37.5) | 0.24 |
| Severe valvular dysfunction, n (%) | 323 (50.4) | 3 (37.5) | 0.47 |
| Vegetation length, mm, mean ± SD | 8.8 ± 7.4 | 8.8 ± 7.7 | 0.47 |
| LVEF, %, mean ± SD | 56.5 ± 9.7 | 53.5 ± 12.0 | 0.58 |
| EuroSCORE II log, mean ± SD | 13.1 ± 16.5 | 13.92 ± 9.7 | 0.65 |
| Surgery, n (%) | 507 (79.1) | 7 (87.5) | 0.62 |
| Cerebral embolism, n (%) | 168 (26.2) | 6 (75.0) | 0.006 |
| Abdominal embolism, n (%) | 85 (13.3) | 1 (12.5) | 0.95 |
| Spondylodiscitis, n (%) | 51 (8.0) | 0 (0) | 0.4 |
| Germ, n (%): | | | 0.92 |
| • Streptococci | 155 (24.2) | 1 (12.5) | |
| • Staphylococci | 180 (28.1) | 2 (25.0) | |
| • Enterococci | 133 (20.7) | 2 (25.0) | |
| • Other | 50 (7.8) | 1 (12.5) | |
| • Negative culture | 122 (19.0) | 2 (25.0) | |

IE = infective endocarditis; BMI = body mass index; LVEF = left ventricular ejection fraction.

**Table 2. Summary of main characteristics of 8 patients with infective endocarditis and coronary embolism.**

| Case # | Age (years) | Valve | Vegetation length (mm), if present | Site of CE, if known | Native/prosthetic valve | Germ type | Surgery | Death |
|---|---|---|---|---|---|---|---|---|
| 1 | 52 | Aortic | - | LCx artery | Biologic | Staphylococcus lugdunensis | Yes | No |
| 2 | 87 | Mitral | 14 | LAD artery | Native | Negative culture | Yes | No |
| 3 | 33 | Aortic | 10 | - | Native | Gemella morbillorum | Yes | No |
| 4 | 76 | Mitral | 20 | Diagonal branch | Biologic | Enterococcus faecalis | Yes | Yes |
| 5 | 77 | Aortic | 12 | RCA | Biologic | Enterococcus faecalis | Yes | No |
| 6 | 43 | Mitral | 6 | LAD artery and OM branch | Native | Staphylococcus aureus | Yes | No |
| 7 | 32 | Aortic | - | RCA | Biologic | Listeria monocytogenes | Excluded | Yes |
| 8 | 25 | Aortic | 14 | - | Mechanical | Negative culture | Yes | No |

LCx = left circumflex; LAD = left anterior descending; RCA = right coronary artery; OM = obtuse marginal.

(a)                                                          (b)

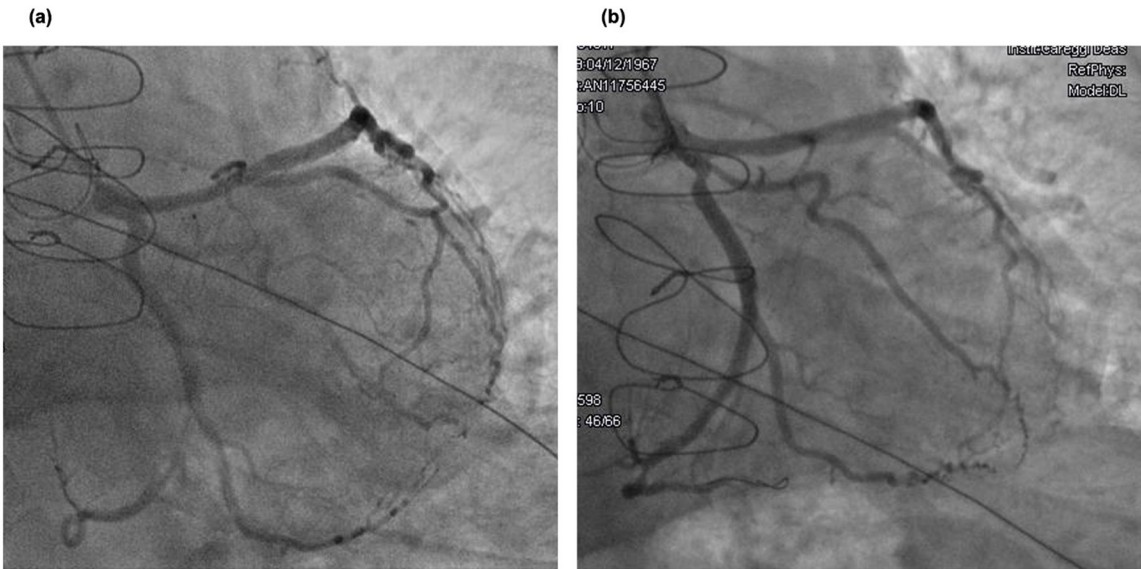

**Fig 1.** Embolic occlusion of left circumflex artery (a) and final angiographic result after thromboaspiration (b) in a patient with infective endocarditis.

stay, the patient complained of dizziness and paraesthesia of the right upper limb. Therefore, brain CT and MRI were performed and showed multiple acute ischemic lesions, the largest located in the left occipital region. She was promptly operated of mitral valve replacement with a biological prosthesis, progressively improved over the next two weeks, and she was finally discharged after in-hospital cardiac rehabilitation.

**Case 3.** A 33-year-old man complained of dyspnoea, asthenia, and fever for about one month. Prior to hospital admission, a chest X-ray and CT scan had resulted negative and fever, but not dyspnoea, had been resolved after empirical antibiotic therapy. He therefore went to a cardiologist who diagnosed severe aortic regurgitation with suspected infective vegetations of aortic valve. Left ventricular distal inferior akinesia was also noted. He was then sent to hospital where a TEE and positron emission tomography confirmed IE of aortic valve. The ECG was normal. The diagnostic workup was completed with a cardiac MRI, showing a moderately dilated left ventricle with distal inferior focal necrosis strongly suggestive of septic embolization. Empirical antibiotic therapy with vancomycin and ceftriaxone was started. Blood cultures were positive for *Gemella morbillorum*. Surgical aortic valve replacement with a biological prosthesis was deemed indicated and the patient was discharged one week later to a cardiologic rehabilitation unit.

**Case 4.** A 76-year-old man who had been operated of surgical mitral valve replacement with a biological prosthesis one year before, was admitted to the emergency department because of paroxysmal nocturnal dyspnoea and abdominal pain for about three days. At blood tests, inflammation markers and D-dimer were elevated. Prompt CT angiography excluded a pulmonary embolism but found a focal splenic embolization. Transthoracic echocardiography and subsequent TEE showed degenerated mitral prosthesis with severe stenosis and mild regurgitation, and vegetations consistent with IE. Antibiotic therapy with ceftriaxone and daptomycin was started. Blood cultures tested positive for *Enterococcus faecalis*. Routine coronary angiography prior to cardiac surgery found an embolic occlusion of first diagonal branch, treated with thromboaspiration. A few hours later he manifested dysarthria, after which a cranial CT scan showed left temporal and parietal hypodense areas. Clinical conditions rapidly

deteriorated and the patient underwent urgent redo mitral valve replacement with a biological prosthesis. The postoperative course was complicated by hemodynamic instability, requiring prolonged treatment with inotropes and vasopressors, and worsening respiratory failure. He died a few weeks later of multi-organ failure.

**Case 5.** A 77-year-old man presented to the emergency department with fever, asthenia, and diffuse thoraco-abdominal pain from one month. 8 years before he had been operated of Bentall procedure due to acute post-traumatic aortic dissection. The ECG showed inferior ST-elevation, and prompt coronary angiography found ostial occlusion of right coronary artery. TEE demonstrated IE vegetations on the aortic valve prosthesis. Empirical antibiotic therapy with daptomycin, gentamycin, and rifampicin was started. A cranial CT revealed an extended right occipital-mesial hypodense area. Blood cultures tested positive for *Enterococcus faecalis*. The patient underwent redo Bentall operation with a biologic prosthesis and coronary artery bypass graft.

**Case 6.** A 43-year-old man with hyperthyroidism and severe hyperbilirubinemia secondary to methimazole toxicity, was repeatedly treated with plasmapheresis via a femoral line. After one week the patient complained of fever and myalgias. Blood cultures and culture exam of the tip of the femoral central line tested positive for methicillin-sensible *Staphylococcus aureus*. Antibiotic therapy with vancomycin and gentamycin was started. After a few days the patient had typical chest pain. The ECG displayed an inferior and posterior ST-elevation myocardial infarction. Immediate coronary angiography showed embolic obstruction of the first obtuse marginal branch, treated with mechanical embolectomy and balloon angioplasty. Culture exam of embolic material tested positive for gram-positive cocci. TTE and TEE demonstrated an IE vegetation on the mitral valve with moderate regurgitation (Fig 2), while a full-body CT scan showed findings of brain micro-embolism. The patient was operated of mitral valve repair with excision of the IE vegetation. The post-operative course was initially complicated by transient hemodynamic instability. At discharge, transthoracic echocardiography showed a well-functioning mitral valve with no residual mitral regurgitation.

**Case 7.** A 32-year-old woman with a history of rheumatic heart disease treated 2 years before with aortic and mitral valve replacement with biological prostheses, was admitted to the hospital due to fever, worsening arthro-myalgias, and purpura of the lower limbs. Blood cultures tested positive for *Listeria monocytogenes*, so antibiotic therapy with meropenem, trimethoprim/sulfamethoxazole and levofloxacin was started. A full-body CT scan displayed an area of detachment at the level of the aortic valve prosthesis with a voluminous periaortic abscess and an ostial obstruction of the right coronary artery. TEE confirmed those findings and, therefore, redo aortic valve surgery was planned. While waiting for surgery, the patient suddenly had right hemiplegia and global aphasia, with CT finding of a large hematoma secondary to ruptured mycotic pseudoaneurysm of the right middle cerebral artery, treated endovascularly (Fig 3). A few days later, she underwent surgical drainage of the intraparenchymal hematoma. Repeated TEE showed mild reduction of the periaortic abscess. Cardiac surgery was therefore deferred, and the patient was transferred to a rehabilitation centre where she suddenly complained of chest pain. The ECG displayed inferior ST- elevation and lateral ST-depression, with echocardiographic findings of interventricular septum and left ventricular inferior wall dyskinesias. Prompt coronary angiography was performed, but after cannulating the left coronary artery the patient had refractory cardiopulmonary arrest and died before the coronary arteries could be visualized.

**Case 8.** A 25-year-old man who had been operated one year before of aortic valve replacement with a mechanical prosthesis for aortic bicuspidia, presented to the emergency department after the sudden onset of typical chest pain following a one-week complain of fever and asthenia, and transient dysarthria. The ECG showed hyperacute T waves and diffuse ST-

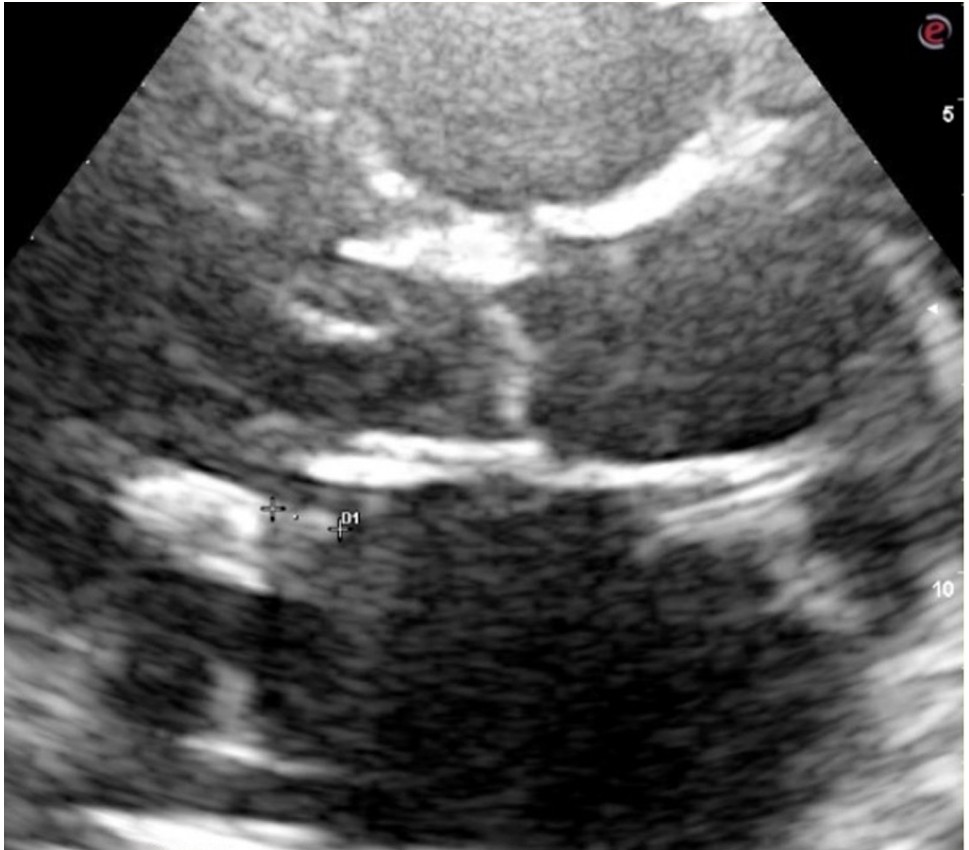

**Fig 2. Endocarditic vegetation on the mitral valve.**

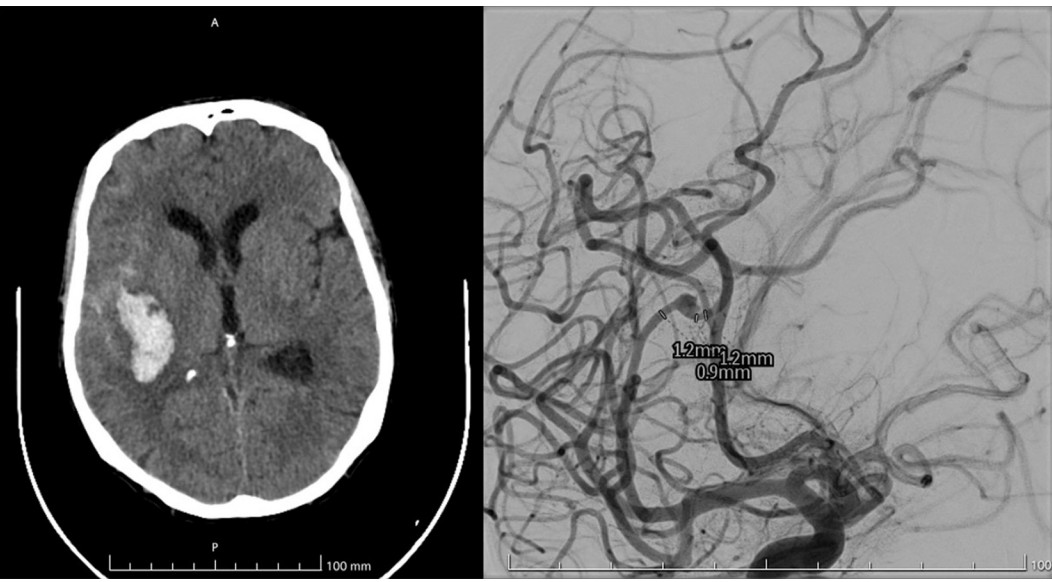

**Fig 3.** Brain CT scan showing a large intraparenchimal hematoma (left) in a patient with angiographical demonstration of ruptured pseudoaneurysm of the right middle cerebral artery (right).

elevation, more pronounced in the lateral leads. Transthoracic echocardiography and subsequent TEE displayed a well-functioning aortic prosthesis, but with an adherent mobile formation on a leaflet. Patient's adherence to oral warfarin was proven by in-range INR values. Blood cultures tested negative. Empirical antibiotic therapy with daptomycin, gentamycin and rifampicin was started. Coronary CT angiography was completely negative, whereas cardiac MRI detected recent transmural infarcted segments in the middle septum and apex, strongly suggestive of microvascular embolization. In the following days, the patient experimented transient dysarthria and diplopia, with MRI finding of multiple embolic lesions of the brain. He underwent urgent cardiac surgery with positioning of an aortic homograft sutured to a Dacron vascular prosthesis. The patient recovered well from surgery without no major complications and at discharge the aortic homograft was well-functioning.

## Discussion

Septic CE is an infrequent, yet recognized, cause of type 2 myocardial infarction [15]. Nevertheless, this event represents a serious complication of IE, with high mortality rates. Cases of CE during nonbacterial endocarditis have also been described [16, 17]. Three studies with more than 500 patients have shown a prevalence of CE in IE of 0.31–0.51% among all cases and of 1.5–3.5% among patients with embolic events. [6, 18, 19] In the setting of ACS without a history of obstructive coronary artery disease, CE must be suspected in the presence of a prosthetic valve associated with fever [6, 20]. Prosthetic valve IE represents one of the major downsides of valvular heart surgery and accounts for 20% of all cases of IE [21]. *Staphylococcus aureus* is currently the most common cause of prosthetic valve IE [22]. *Staphylococcus lugdunensis* affects mostly native valves and has an aggressive course; in prosthetic valve IE, it is usually localized in the aortic position and associated with abscess formation and poor prognosis [23]. Perivalvular abscess formation is associated with a more aggressive course and worst outcomes; it is more commonly seen as a complication of prosthetic valve IE, especially aortic, and appears to occur more often in IE caused by *Staphylococcus aureus* [24, 25].

Systemic embolization is one of the most dreaded complications of IE, occurring in 22–50% of cases [2, 26]. Brain, spleen and kidney are the most frequent sites of embolism for left-sided IE, while pulmonary embolism is frequent in right-sided IE. Stroke may be the first clinical manifestation of IE and represents a complication associated with increased mortality and morbidity [1, 2, 27].

Studies investigating specific risk factors for CE are scarce [28]. A study showed that CE was most commonly seen in patients with aortic valve endocarditis, and this may be explained by the proximity of aortic vegetations to the coronary ostia [29].

CE is most often secondary to arterial thromboembolism, mainly in the setting of atrial fibrillation; septic emboli are less frequent, complicating <1% of IE [25, 30, 31].

ACS as a complication of IE can also occur due to compression of the coronary artery ostia by an aortic root abscess [30, 32]. Presence of prosthetic valve, a new precordial murmur, fever and leucocytosis are all factors compatible with CE as the potential cause of ACS but are not enough to exclude a priori atherosclerotic cardiovascular disease [33]. CE is most frequent in the left anterior descending artery, due a facilitating anatomy [33, 34–36]. In a study by Nazir et al, who described 100 patients with ST-elevation myocardial infarction associated with IE reported by 95 articles, embolic events occurred mainly in the left coronary branches (87% of cases: 75% involving the left anterior descending artery, 10% the left circumflex artery, 2% the left main) while the right coronary artery was the culprit vessel in the remaining cases [32]. Embolization of the left circumflex artery, as described in our #1 and #5 cases, is therefore an atypical clinical scenario.

Besides the management of possible complications, the treatment mainstay of IE remains appropriate antibiotic therapy and surgery. Treatment of CE secondary to IE is challenging. Embolectomy, balloon angioplasty, and stent placement may be required [37, 38]. Aspiration and/or mechanical thrombectomy has been advocated as a reasonable first-line choice in this context, particularly considering the risks of angioplasty and stenting of an infected thrombus [32, 39–42]. In fact, stent placement in septic CE cases is still debated due to the risk of dilation site, mycotic aneurysm formation and rupture, and distal embolization [20, 37, 43]. Yet, mycotic aneurysm formation may be a serious complication of balloon angioplasty too, due to potential intimal disruption and secondary bacterial seeding [44, 45]. Indeed, in our case series balloon angioplasty was deemed to be necessary in only one case, and in no case a stent was placed to optimize results obtained with aspiration or mechanical thrombectomy. Lastly, there is an absolute contraindication to fibrinolytic therapy in the setting of IE due to a high-risk of intracranial haemorrhage [46, 47].

Since evidence of ACS complicating IE is derived mainly from case reports, uncertainty remains regarding the best treatment strategy. In conclusion, an early invasive catheter-intervention appears a feasible option, while conservative, or multi-step, management is associated with worst outcomes, [20, 32, 33, 36, 48, 49] as demonstrated by our #7 case. Our retrospective analysis of 649 patients with IE has demonstrated a 1.2% prevalence of CE. We have found a statistically significant association between coronary and cerebral embolism, with a post-hoc power of 88.1%. Given the rarity of CE, no other statistically significant difference in either baseline characteristics or mortality rates was found. To the best of our knowledge, the association rate between CE and cerebral embolism in patients with IE has never been described in the literature. Therefore, septic cerebral embolism could represent a risk factor for CE. The typical clinical presentation of CE during IE is ACS with chest pain, ECG changes and troponin elevation. However, in case #3 and #4, CE has passed asymptomatically. In case #3, the diagnosis was made by MRI. In case #4, coronary angiography was performed as routine before cardiac surgery, and the finding of CE was incidental. Of the 8 patients with CE, 3 were treated with thromboaspiration and 7 underwent cardiac surgery (see Table 2). In our case study, two patients' blood cultures were negative, and two patients tested positive for *Enterococcus faecalis* (see Table 2). No other specific microorganism was prevalent. These findings are likely attributable to the low number of patients with CE in our population. Vegetation length was greater than 1 cm in 5 of our 8 patients with CE, which may explain the high tendency of these vegetations to embolize (see Table 1). Of the 8 patients with CE, 5 had aortic valve and 3 had mitral valve IE. Only one patient had a mechanical prosthetic valve, 4 had a biologic prosthetic valve and 3 had no prosthesis (see Table 2). The coronary involvement was heterogeneous. Two patients (25%) died close to the infectious and embolic events. This finding might suggest how IE complicated by CE represents a serious clinical condition with high morbidity and mortality that should be promptly recognized and treated.

Last but not least, it is important to emphasize how aggressive surgical management of complicated IE can improve patients' prognosis. In fact, it has been demonstrated that conservative approach was associated with an adverse prognosis [50]. It has also been found that vegetations with a length > 10 mm increases the risk of embolic events [51]; early surgical treatment could therefore avert these serious complications.

## Conclusion

CE is a rare and serious complication of IE. In our case series, we have found a significant association between CE and cerebral embolism. There were no association with the presence of native or prosthetic valve. The mortality rate was high (25%). Due to the low prevalence,

multicentre studies including larger numbers of patients are needed to be able to identify risk factors of CE in IE that will help in the early recognition of this condition so that clinicians can have a crucial impact in improving their patients' prognosis.

## Limitations

Our study has several limitations. The number of patients with IE and CE is small and the cases included in this study are drawn from a single centre, therefore this limits the ability to generalize findings. Multicentre studies are therefore needed to confirm our results. The number of patients with CE could be higher than what we reported. In fact, besides symptomatic patients or patients with ECG, echocardiographic or MRI signs of myocardial ischemia or infarction, only patients with IE and an indication to surgery underwent coronary angiography. Thus, patients with IE and not candidates for surgery could have developed CE that passed asymptomatic.

## Author Contributions

**Conceptualization:** Pierluigi Stefàno.

**Data curation:** Valentina Scheggi.

**Formal analysis:** Niccolò Marchionni.

**Investigation:** Ruggero Mazzotta, Matteo Orlandi.

**Methodology:** Valentina Scheggi.

**Supervision:** Niccolò Marchionni, Pierluigi Stefàno.

**Writing – original draft:** Ruggero Mazzotta, Matteo Orlandi.

**Writing – review & editing:** Valentina Scheggi.

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
