## [Decision Letter · Decision Letter 0]

16 Oct 2024

PONE-D-24-38850Coronary embolism in left-sided infective endocarditis. A retrospective analysis from a high-volume surgical centre and review of the literaturePLOS ONE

Dear Dr. Orlandi,

Thank you for submitting your manuscript to PLOS ONE. After careful consideration, we feel that it has merit but does not fully meet PLOS ONE’s publication criteria as it currently stands. Therefore, we invite you to submit a revised version of the manuscript that addresses the points raised during the review process.

**Dear authors,**

**I'm pleased to inform you that your manuscript has been reviewed, and I believe it has great potential.****The reviewers have provided insightful comments and suggestions. Please address these minor revisions to enhance the quality of your paper. I am confident that your responses will significantly strengthen your work.** Please submit your revised manuscript by Nov 30 2024 11:59PM. If you will need more time than this to complete your revisions, please reply to this message or contact the journal office at plosone@plos.org. Please include the following items when submitting your revised manuscript:A rebuttal letter that responds to each point raised by the academic editor and reviewer(s). You should upload this letter as a separate file labeled 'Response to Reviewers'.A marked-up copy of your manuscript that highlights changes made to the original version. You should upload this as a separate file labeled 'Revised Manuscript with Track Changes'.An unmarked version of your revised paper without tracked changes. You should upload this as a separate file labeled 'Manuscript'.If applicable, we recommend that you deposit your laboratory protocols in protocols.io to enhance the reproducibility of your results. Protocols.io assigns your protocol its own identifier (DOI) so that it can be cited independently in the future. For instructions see: https://journals.plos.org/plosone/s/submission-guidelines#loc-laboratory-protocols. Additionally, PLOS ONE offers an option for publishing peer-reviewed Lab Protocol articles, which describe protocols hosted on protocols.io. Read more information on sharing protocols at https://plos.org/protocols?utm_medium=editorial-email&utm_source=authorletters&utm_campaign=protocols.

We look forward to receiving your revised manuscript.

Kind regards,

Ahmed Qasim Mohammed Alhatemi

Academic Editor

PLOS ONE

Journal Requirements:

Reviewers' comments:

Reviewer's Responses to Questions

**Comments to the Author**

1. Is the manuscript technically sound, and do the data support the conclusions?

Reviewer #1: Yes

Reviewer #2: Yes

Reviewer #3: Yes

2. Has the statistical analysis been performed appropriately and rigorously? 

Reviewer #1: I Don't Know

Reviewer #2: Yes

Reviewer #3: Yes

3. Have the authors made all data underlying the findings in their manuscript fully available?

Reviewer #1: Yes

Reviewer #2: Yes

Reviewer #3: Yes

4. Is the manuscript presented in an intelligible fashion and written in standard English?

Reviewer #1: Yes

Reviewer #2: Yes

Reviewer #3: Yes

5. Review Comments to the Author

Reviewer #1: Embolic events in infective endocarditis remain a major concern affects management.

1. Being a retrospective review, is there any chance that some coronary embolization were not diagnosed because they did not result in clear ACS? was there any biomarker data that may shed more light on this point?

2. How systematic was the search for cerebral embolization, again could many cerebral emboli have been missed?

3.The cases document when the coronary events occurred and I think it would be useful to include the time from presentation to the occurrence of coronary embolism in hours and also the duration of antibiotic therapy in tables 1 and 2 to see if embolic events represent delays in initiating antimicrobial therapy in relation to the diagnosis of possible infective endocarditis and try to understand better the predictors of coronary embolization.

4. It is not clear from the manuscript whether vegetation length was by 2D or 3 D. This needs to be clarified.

5. Also I did not see any specific analysis for Coronary Embolization related to the mitral valve.

6. Is there any other echo parameter of the vegetation that can be related to cerebral or coronary embolization.

Answers to all the above points should be incorporated into the manuscript so that readers may benefit more from this case series.

7. A limitation section need to be included.

Reviewer #2: Thank you for this case series. I would like the authors to clarify how they decided to deal with the coronary emboli. In some cases, they left it alone and in some cases, they intervened. On what basis was the decision made? I also want to know if they analysed the recovered embolic material and identified the causative pathogen, especially in culture negative cases. Finally, in case no.3, the authors state that the empirical antibiotics used were vancomycin and ceftriaxone, this is not a regular combination, why was it used?

Reviewer #3: The authors presented a compilation of rare cases coronary embolism caused by infective endocarditis. The clinical history of each case was also summarised well. This is not an easy task to collect these types of rare cases. The efforts by the author should be commended.

The possible association between coronary embolism and cerebral embolism was also noted in the article. Future studies should look into this association. If the association is confirmed patients with infective endocarditis should have investigated for both complications once they develop one.

Overall this article is good and can be accepted for publication.

6. PLOS authors have the option to publish the peer review history of their article (what does this mean?). If published, this will include your full peer review and any attached files.

Reviewer #1: **Yes: **Shukri AlSaif

Reviewer #2: **Yes: **Ahmed A. Elamragy

Reviewer #3: No

---

## [Author Response · Author response to Decision Letter 0]

12 Nov 2024

Responses to Reviewer 1

Q1. Being a retrospective review, is there any chance that some coronary embolization were not diagnosed because they did not result in clear ACS? was there any biomarker data that may shed more light on this point?

R1. We thank the reviewer for this comment. All patients with infective endocarditis included in our study were routinely screened with ECG and echocardiogram during their hospitalization. So, even if we cannot exclude that coronary embolization could have passed asymptomatic, we think that the proportion of patients correctly diagnosed with coronary embolization because of signs and symptoms of ischemia significantly outweighs the number of cases not diagnosed. Moreover, the vast majority of patients with left-sided infective endocarditis were treated surgically in our center and, thus, were screened with CT or invasive coronary angiography before surgery. Myocardial enzymes were drawn in all patients treated surgically, but we have to consider that acute myocardial injury is very frequent in the post-operative course and therefore cardiac enzymes values were not specific for the diagnosis of coronary embolization. After your suggestion, we further clarified this point in the “Limitations” section of our manuscript.

Q2. How systematic was the search for cerebral embolization, again could many cerebral emboli have been missed?

R2. In our center all patients with left-sided infective endocarditis are routinely screened with brain CT to search for potential cerebral emboli. Therefore, we think that only a slight minority of cerebral emboli could have been missed. In patients with signs or symptoms of cerebral ischemia, repeated CT scans or MRI scans were performed, reducing the possibility of missing the diagnosis of cerebral embolization. 

Q3. The cases document when the coronary events occurred and I think it would be useful to include the time from presentation to the occurrence of coronary embolism in hours and also the duration of antibiotic therapy in tables 1 and 2 to see if embolic events represent delays in initiating antimicrobial therapy in relation to the diagnosis of possible infective endocarditis and try to understand better the predictors of coronary embolization.

R3. We thank the reviewer for this comment. Given the fact that our center is a third-level hospital for the treatment of infective endocarditis, especially because of the presence of in-site cardiac surgery facilities, many patients were transferred from other hospitals after the diagnosis of infective endocarditis was made and when antibiotic therapy was already started. Unfortunately, data regarding the time from presentation to the occurrence of coronary embolism and the duration of antibiotic therapy cannot be retrieved from our electronic charts. We agree with the reviewer that including these data could have shed new insights on the knowledge of the predictors of coronary embolization. This could be the focus of future studies.

Q4. It is not clear from the manuscript whether vegetation length was by 2D or 3D. This needs to be clarified.

R4. Vegetation length was measured with 2D transesophageal echocardiography.

Q5. Also I did not see any specific analysis for Coronary Embolization related to the mitral valve.

R5. We thank the reviewer for this comment. In our study only patients with left-sided infective endocarditis were included. We reported in Table 1 the rates of aortic valve infective endocarditis (56.5%) and aortic valve involvement among patients with coronary embolism (62.5%). Mitral valve infective endocarditis was present in 43.5% of patients and mitral valve was involved in 37.5% of patients with coronary embolism. The association between coronary embolism and aortic or mitral valve infective endocarditis was not statistically significant (p= 0.86). Involvement of anterior or posterior mitral leaflet, presence of mitral stenosis or mitral regurgitation and grading of valve disease were not parameters included in our dataset and therefore no specific analysis in terms of the relationship with coronary embolism can be done.

Q6. Is there any other echo parameter of the vegetation that can be related to cerebral or coronary embolization.

R6. In our analysis we did not find a significant association between vegetation length and the risk of coronary embolization. Other echo parameter, such as vegetation mobility and site of attachment, especially in the case of the mitral valve, could be potentially associated to cerebral or coronary embolization. However, these data were not routinely collected and therefore, in our study, no statistical analysis could be performed.

Q7. A limitation section need to be included.

R7. We thank the reviewer for the suggestion. We added a “Limitations” section to our manuscript.

Response to Reviewer 2 

Thank you for this case series. I would like the authors to clarify how they decided to deal with the coronary emboli. In some cases, they left it alone and in some cases, they intervened. On what basis was the decision made? I also want to know if they analyzed the recovered embolic material and identified the causative pathogen, especially in culture negative cases. Finally, in case no.3, the authors state that the empirical antibiotics used were vancomycin and ceftriaxone, this is not a regular combination, why was it used?

R. As we outlined in the discussion, there is no clear evidence of the best strategy of treatment for patients with infective endocarditis and coronary embolization. In the majority of cases presenting with acute coronary syndrome, after the diagnosis of coronary embolization thromboaspiration was performed (case #1, #4, #6). In one case (case #5), coronary artery bypass graft was preferred. In one case (case #2) no treatment was performed, because of the distal localization of the occlusion. The patient of the case #7 died before that coronary angiography could be performed. In case #3 and #8, the diagnosis of coronary embolization was made after the results of the MRI, so that an interventional procedure could not be performed. In only one case (case #6) the embolic material was retrieved, and cultural exams tested positive for the same bacteria identified with blood cultures. We recognize that routine cultural exams on embolic material could have helped in the diagnosis of patient with negative cultures (case #2). In case #3, empirical antibiotic therapy with vancomycin and ceftriaxone was started. We recognize that this combination therapy is not usual, but the patient was transferred from another center after this combination of antibiotics was started. Treatment was later shifted to ceftriaxone and gentamicin after that Gemella morbillorum was found on the valve samples.

---

## [Editor Report · Decision Letter 1]

15 Nov 2024

Coronary embolism in left-sided infective endocarditis. A retrospective analysis from a high-volume surgical centre and review of the literature

PONE-D-24-38850R1

Dear Dr. Orlandi,

We’re pleased to inform you that your manuscript has been judged scientifically suitable for publication and will be formally accepted for publication once it meets all outstanding technical requirements.

Kind regards,

Ahmed Qasim Mohammed Alhatemi

Academic Editor

PLOS ONE
---

## [Editor Report · Acceptance letter]

21 Nov 2024

PONE-D-24-38850R1 

PLOS ONE

Dear Dr. Orlandi, 

I'm pleased to inform you that your manuscript has been deemed suitable for publication in PLOS ONE. Congratulations! Your manuscript is now being handed over to our production team.

Kind regards, 

on behalf of

Dr. Ahmed Qasim Mohammed Alhatemi 

Academic Editor

PLOS ONE